# Effectiveness and Safety of COVID-19 Vaccination in Patients with Malignant Disease

**DOI:** 10.3390/vaccines11020486

**Published:** 2023-02-20

**Authors:** Li Zhao, Lin Fu, Yuqin He, Han Li, Yixuan Song, Shaoyan Liu

**Affiliations:** 1Department of Head and Neck Surgery, National Cancer Center/National Clinical Research Center for Cancer/Cancer Hospital, Chinese Academy of Medical Sciences and Peking Union Medical College, Beijing 100021, China; 2Institute of Plastic Surgery, Chinese Academy of Medical Sciences and Peking Union Medical College, Beijing 100144, China

**Keywords:** COVID-19, SARS-CoV-2 variants, vaccines, cancer

## Abstract

A novel virus named SARS-CoV-2 has caused a worldwide pandemic, resulting in a disastrous impact to the public health since 2019. The disease is much more lethal among patients with malignant disease. Vaccination plays an important role in the prevention of infection and subsequent severe COVID-19. However, the efficacy and safety of vaccines for cancer patients needs further investigation. Encouragingly, there have been important findings deduced from research so far. In this review, an overview of the immunogenicity, effectiveness, and safeness of COVID-19 vaccines in patients with cancer to date is to be shown. We also highlight important questions to consider and directions that could be followed in future research.

## 1. Introduction

The emergence of a novel species of coronavirus in 2019, named severe acute respiratory syndrome coronavirus 2 (SARS-CoV-2) by WHO [1], caused a worldwide pandemic with disastrous impacts to public health worldwide. The mortality rate for the disease is higher in patients with certain comorbidities, including cancer [2,3,4]. Even though patients with malignant disease overlap risk factors with the general population, such as advanced age, malignancy itself may contribute as an independent risk factor [5,6,7]. The risk of severe COVID-19 is obviously higher in patients <65 years of age with malignant disease [6,8]. It is considered that malignancy is associated with a significant increase in case fatality for both hematologic and solid malignancies [9]. Mortality rates among patients with active cancers are significantly higher, affected by their compromised immunity [8]. Additionally, mortality rates can also be influenced indirectly. Disruptions to the healthcare system caused by the pandemic can result in cancer treatment delays, frequent treatment modifications, and reduced screening, which impacts cancer-specific survival [10,11]. The incidence of long-term COVID-19 sequelae also affects up to 15% of cancer patients and adversely affects the survival rate and tumor prognosis after COVID-19 recovery [11]. However, it is also important to note that mortality rates in these studies were usually derived from inpatient data, so the specific rates might be overestimated.

Vaccination plays an important role in the prevention of infection and subsequent severe COVID-19 among the general population (Figure 1). As cancer patients face a higher risk of mortality from COVID-19, effective preventative efforts are especially important in this population, with vaccination being one of the most effective methods. It has already been proved that vaccination is a safe and powerful tool to protect this population against other infectious diseases [12,13,14]. Previous research also indicates that patients with cancer benefit from additional vaccine doses [15], which is in line with the current concept of prime–boost vaccination. However, insufficient data were collected to directly ascertain the efficacy and safety of vaccines that are currently available for COVID-19 patients with cancer. Compared to the general population, patients with malignancy are more likely to develop a less proficient immune response upon vaccination. This is mainly caused by disease-associated or therapy-led immune deficiency. Therefore, patients with cancer are usually prioritized for vaccinations but excluded from registration in clinical trials [16,17]. This means that vaccine efficacy in this population has had to be indirectly deduced from an immunological or antibody-response perspective [18,19,20].

Patients with malignant disease have a higher mortality rate from SARS-CoV-2. Therefore, it is vital that we understand how we can more effectively prevent infection or reduce disease severity through vaccination. This review will provide an overview of the current knowledge of COVID-19 vaccines in patients with malignant disease, including vaccine effectiveness, factors that could negatively influence immune response, and vaccine toxicity and safety. We tried to address questions based on available data. Information mentioned in this review may provide us with a better perspective to guide our clinical recommendations and future vaccination research programs.

During the SARS-CoV-2 infection process, viral antigens such as the spike (S) protein are recognized and presented by antigen-presenting cells. Vaccines (such as mRNA and adenoviral vector vaccines) mimic this response by encoding the spike protein. Antigen-presenting cells recognize the spike protein and present it to effector cells. This results in the activation of B cell responses, which is responsible for providing humoral immunity and the activation of effector cells, including T helper CD4+ (Th cell) cell and CD8+ T cell, mediating cellular immunity. Memory T and B cells persist in the periphery and can expand in response to secondary exposure. This figure was created with BioRender.

## 2. Literature Search

In this review, we used different electronic databases including PubMed, EMBASE, and the first 20 pages of the Google Scholar. The time scale was from 15 December 2019, to 15 October 2022. The language of the articles that were reviewed was limited to English. Information retrieval used the combinations of the following medical subject headings and key words: COVID-19, SARS-CoV-2, variants, alpha, beta, gamma, delta, iota, epsilon, vaccine, safety, efficacy, side effects, effectiveness, clinical trial, observational study, randomized controlled study, vaccination, mRNA vaccine, adenovirus vector vaccine, subunit vaccine, inactivated vaccine, ChAdOx1 nCoV-19, Ad26.COV2.S, mRNA-1273, BNT162b1, BNT162b2, rAd26, rAd5, MF59-adjuvanted spike glycoprotein-clamp, immunogenicity, immunotherapy, neoplasms, cancer, hematologic neoplasms, lymphoma, leukemia, multiple myeloma, myelodysplastic syndromes, myelofibrosis, B-Lymphocytes, T-Lymphocytes. Studies were screened and we included relatively larger studies that described the immunogenicity, efficacy, and safety of COVID-19 vaccines in patients with cancer. Studies were excluded with relatively small cohorts, except for areas that have limited articles.

## 3. Vaccine Effectiveness in Patients with Malignant Disease

Clinical vaccine efficacy, which is usually defined as the prevention of symptomatic COVID-19, is generally less proficient in patients with malignant disease [20,21,22,23,24,25,26,27,28,29]. The evaluation of vaccine-induced immune response in cancer patients is mainly focused on assessing the humoral immunity response. Humoral immunity response is primarily evaluated by establishing seroconversion rates and mean anti-body titers of antibodies that are spike-reactive or RBD-reactive [30,31,32,33]. Cellular immunity, mainly addressed by T cell response, is relatively limited and often provided along with humoral immunity.

According to a large US Veterans study including cancer patients who received mRNA vaccines, the overall vaccine effectiveness (VE) in patients with cancer was about 58%, which is lower compared to the general population [29]. The high risk of VE reduction against COVID-19 experienced by cancer patients has also been observed in several other studies [18,20,21,27,31,34,35]. A UK prospective cohort study of 6.9 million vaccinated participants identified hematological cancer (HR 1.86), respiratory tract cancer (HR 1.35), receiving chemotherapy (HR 3.63–4.3), and receiving bone marrow or solid organ transplantation within the past 6 months (HR 2.5) as risk factors for COVID-19-related death, despite receiving two doses of vaccination [36]. Fortunately, some research also indicates that in long-term follow ups, certain cancer survivors can develop higher VE, closer to the rates of the general population [29,37]. 

### 3.1. Humoral Immunity: Serology and Neutralizing Antibodies

#### 3.1.1. Effectiveness of Vaccination on Humoral Immunity

Humoral immunity is mostly evaluated through the rates of seroconversion and mean antibody titers. According to a prospective observational study of the mRNA vaccine BNT162b2 (Pfizer-BioNTech), after receiving one does of the vaccine, the proportion of participants with positive anti-S IgG titers were 38% and <20% in patients with solid tumors and hematological malignancies, respectively, compared with 94% in the control cohort [20]. Consistent with the general population, neutralizing responses to variants of concern (VOCs) decrease progressively in patients with cancer [18,38]. In a predictive model study, neutralization titers against some SARS-CoV-2 variants of concern were reduced compared with the vaccine strain [39]. This might be the reason why, in patients with hematological malignancies, only 31% patients had detectable titers with activity against the Delta variant after two vaccine doses, compared to 56% against the previous Wuhan strain [18]. 

#### 3.1.2. Risk Factors for Poor Humoral Immune Response to Vaccination

##### Solid Cancer

The presence of seroconversion can be detected in most patients (>90%) with solid tumors, which is comparable with the general population [18,34,40]. However, the specific conversion rate tends to be lower, as it is impacted by impaired immunogenicity [20,25,35]. A meta-analysis reported that 297 patients with cancer who have completed their vaccine regiment had a lower seroconversion rate compared with the 140-participant control group (RR 0.95; 95% CI 0.92–0.99) [26]. 

The cancer population and the general population share common risk factors for reduced seroconversion: age, sex, and the type of vaccine [18,30,34] (adenovirus-vectored vaccines are less effective compared to mRNA vaccines). The CAPTURE study compared the use of BNT162b2 (mRNA) and AZD1222 (adenovirus-vectored) in patients with malignant disease. In comparison with the AZD1222 cohort, an exceptionally higher proportion of the BNT162b2 cohort developed neutralizing antibodies (nAbs) against the VOCs, along with significantly higher median nAb titers [18]. Similar results have been reported by Astha et al. [34]. Thus, it might be important to prioritize mRNA vaccines for cancer patients wherever possible.

Specific cancer therapies can impair vaccine-induced immunogenicity. Chemotherapy: patients who received chemotherapy within 3 months prior to the first vaccination dose were estimated to have a vaccine effectiveness of 57% (95% CI, –23% to 90%) starting 14 days after the second dose vs 76% (95% CI, 50% to 91%) for those receiving endocrine therapy and 85% (95% CI, 29% to 100%) for those who had not received systemic therapy for at least 6 months prior [29]. Several other studies also identified chemotherapy as a risk factor for lower seroconversion and neutralizing responses [23,30,31,35,41,42,43,44]. Notably, seroconversion may not be affected by the timing of vaccination of ongoing chemotherapy cycles [31,34]. Therefore, centers do not have to reschedule chemotherapy plans for vaccination. However, to avoid acute adverse effects, vaccines should still not be administered with chemotherapy on the same day. Immune checkpoint inhibitors: the evidence is mixed on the impact of inhibitors on vaccination. In a recent study, 7% (9/131) of the patients treated with immunotherapy were classified as suboptimal responders or non-responders, while the extent of seroconversion is generally high [40]. A study of breast cancer patients receiving CDK4/6 inhibitors reported no significant decrement in response to the first dose of COVID-19 vaccines. However, another study showed significantly lower titer after vaccination in patients receiving CDK4/6 inhibitor therapy [34]. This is in line with the result of another study, where up to day 30 after the second dose, ovarian cancer patients receiving PARP inhibitors had significantly lower nAbs in comparison to matched healthy volunteers [45]. Steroids: a study reported chronic steroid as an independent risk factor for reduced seroconversion in solid cancer therapy [46]. 

##### Hematological Malignancies

An exceptionally lower seroconversion rate has been observed in patients with hematologic malignancies compared to those with solid tumors after complete immunization (65% vs. 94%; *p* < 0.0001) [26]. This is in line with another study that compares seropositive rate after vaccination in this population against the comparison group (75% vs. 99%; *p* < 0.001) [41]. However, vaccination responses are similar to the general population in long-term hematological malignancy patients, including patients that received very immunosuppressive therapies [29,37]. Patient-specific risk factors include sex [47,48,49], age [47,48,49,50,51], and type of vaccine [19,52]. A superiority in VE of mRNA vaccines has also been observed in patients with Hematological Malignancies [18]. Additionally, interestingly, within the mRNA vaccines, mRNA-1273 is more effective than BNT162b2 [28,50]. Another risk factor is progressive disease [50,53,54,55] (and other factors that result in the disorder of the immune system, such as reduced levels of uninvolved immunoglobulins or lymphopenia) [41,48,50,55,56,57,58]. Patients with different types of hematological malignancies may also respond differently to the same COVID-19 vaccine. A study shows that among patients with chronic lymphocytic leukemia (CLL), non-Hodgkin’s lymphoma (NHL), multiple myeloma (MM), chronic myeloid leukemia (CML), myeloproliferative neoplasms (MPN), and myelodysplastic syndromes (MDS), patients with CLL, NHL, and MM had the lowest seropositivity rates [35]. Therefore, it might be important to run subgroup analyzes in future research. Specific therapies can also impair vaccine-induced immunogenicity. Antibody responses were substantially reduced in patients receiving anti-CD20 antibody therapies [27,31,41,47,48,51,54,58,59,60,61,62,63,64,65,66,67], BTK inhibitors [41,48,51,54,56,59,61,62,65], BCL inhibitors (such as venetoclax) [41,48,51,61,62,63,64,65], BCMA-targeted therapies [57,62,68,69], anti-CD38 antibody therapies [57,62,67,68,69,70], and JAK inhibitors [41,59]. Over 50% of patients exhibited less seropositive responses compared to control groups, especially in patients undergoing active treatment. Depletion of normal B cells is the main reason for negative seroconversion-receiving anti-CD20 antibody therapies, BTK inhibitors, or BCL inhibitors. Treatment with B cell–directed therapies may reduce the production of vaccination-induced antibodies in patients with hematological malignancy because of B cell depletion and/or disruption of the B cell-receptor signaling pathway. A duration of time before vaccination from the last B cell–directed treatment may result in improved antibody titers [64]. Additionally, **steroids**, especially active steroid therapy, seems to have a negative effect on humoral response [47,49,50,71]. Compared to specific therapies mentioned above, the negative effect seems to be milder, probably dosage-dependent, influencing <50% of patients with hematological malignancy. However, the negative effect of TKIs is rarely seen [41,59]. Encouragingly, it seems that deficient humoral immunity can be improved through the treatment of the primary disease, according to a study in which 79.2% of patients reached a positive serological response after receiving effective treatment. [48]. This is in line with the observation of long-term hematological malignancy patients, whose VE are comparable with the general population, including patients that received very immunosuppressive therapies [29,37].

### 3.2. Cellular Immunity: T Cell Responses

#### 3.2.1. Effectiveness of Vaccination on Cellular Immunity

The effectiveness of vaccination on cellular immunity can be measured by flow cytometric analysis of cellular activation-induced markers [72], IFN-γ release [18,40,54], or combined IFN-γ and IL-2 release [20,66]. Waned T cell response could be detected more clearly with IFN-γ release or cellular activation-induced markers [18,32]. Cellular immunity is often weaker in cancer patients compared to the general population. Only 46–79% T cell responses are detected in solid tumor patients according to studies above. However, several studies still show comparable responses between the cancer population (both solid cancer and hematological patients) and the general population [18,32,33,54,72]. Additionally, strong T cell responses can be elicited after stimulation with SARS-CoV-2-derived peptides [34,73,74,75,76]. In contrast to humoral immunity, cellular immunity may correlate better with long-lasting immune memory and protection from severe disease [77]. T cell response is less affected by mutations, and epitopes are more broadly conserved [78]. Therefore, T cell responses are more robust in patients with hematological malignancies; they can be detected in 34% to 75% of patients in whom serological response is negative (although 34% of seronegative individuals had CD4 responses with mainly IL-2-only monofunctional cells) [18,54,58,68].

#### 3.2.2. Risk Factors for Poor Cellular Immune Response to Vaccination

Criteria for T cell positivity alters among studies. Some studies use the presence of activation markers on virus-specific T cells, while others have quantified cellular cytokine secretion. Therefore, the establishing of risk factors for poor cellular immune responses to vaccination is more challenging than for humoral immunity. 

##### Solid Cancer

Limited data are available on the risk factors of cellular immunity in solid cancer. In a study including cancer patients receiving BNT162b2 mRNA, chemotherapy or steroids within 15 days of vaccination were associated with reduced T cell responses to vaccination [33]. Other studies also reported reduced T cell responses but did not specify treatment [20,72]. Interestingly, T cells are detectable in the absence of antibody responses in patients receiving chemotherapy or immune checkpoint inhibitors [40]. 

##### Hematological Malignancies

Compared to the humoral response to vaccination, specific therapies (especially B cell-depleting therapies as anti-CD20 antibody therapies) seem to have a weaker impact on T cell responses [18,47,58,66]. CD8+ responses have been detected in hematologic cancer patients that received B cell-depleting therapies, even in the absence of humoral responses [79,80]. HSCT and allogeneic-HCT: recipients of HSCT and allogeneic-HCT have a much weaker T cell response; while serological response is detectable, T cell responses seem to occur in only 20–30% of patients [81,82]. Moreover, T cell performance might be influenced by the specific type of the disease. CML patients may achieve a polyfunctional T cell rate of 12/15 (80%) [83], while 80% and 60% of MPN patients showed a polyfunctional response of CD4 and CD8 T cells, respectively [84]. However, seronegative MM patients had significantly reduced CD4 T cell responses compared to those of healthy controls. Spike stimulation increases IL-2 levels in patients with hematological malignancies, but the IFN-γ and tumor necrosis factor α (TNF-α) remain lower, which indicates a reduced magnitude of protection by T cell response [68]. Other studies using receptor-binding domain (RBD) as stimulation still elicited weaker immunogenicity responses in patients with MM vs healthy controls (34.2% vs. 71.4%). Moreover, in other hematological patients, low-RBD-specific T cell responses can also be seen [20,33]. This probably means that instead of disease-specific immunosuppression, cellular immunity is linked to the failure to generate a durable immune response to novel antigens. Humoral and cellular responses were often found to be discordant in patients with cancer. Thus, the disharmony of the humoral and cellular immune system is a possible cause of poor vaccine response. 

### 3.3. Prime–Boost Vaccination: Solution for Waning Immunity and VOCs

Vaccine efficacy reduces over time due to the waning of humoral immunity and the emergence of novel VOCs. In line with reports from the general population, neutralizing responses to VOCs decrease progressively in patients with malignant disease [18,38]. However, the combination effect of VOCs and malignancy can lead to drastically reduced VE, which is more likely be seen in patients with hematological malignancies [39]. It has been reported the VE against the Delta variant was only 31%, compared to the previous 56% VE against the previous Wuhan strain [18]. Encouragingly, booster vaccination, as a solution for waning immunity and VOCs, is well tolerated in cancer patients [32]. Several studies have shown the poor immune effect of a single-dose vaccination, and that it can be significantly boosted through a second dose, with seropositivity boosted up to 75% and 95% after the second dose [22,25,30,31,32,34,35,44,46,54,60,72,85,86,87,88]. Therefore, booster doses for malignancy patients were urgently rolled out to compete with the waning humoral immunity. Increased performance of humoral immunity, measured by antibody titers, in patients with cancer have been observed, and neutralizing antibodies (nAbs) have also shown threefold increases compared with pre-booster doses [27,32,52,89,90,91,92,93].

Receiving treatment does not appear to be a contraindication for booster vaccination, as it increases the antibody response in patients with solid tumors, even in those receiving active treatment (intravenous anticancer medication) [94]. Furthermore, booster vaccination induced seroconversion in previously seronegative patients; up to 56% patients showed a humoral response after the second vaccine dose [76]. 

In certain circumstances, a positive immune response might not be achieved with two doses, especially to avoid response decrement due to novel VOCs. A third dose might be necessary. The percentage of patients with detectable neutralizing responses to VOCs can be broadened following booster vaccination; participants who had low neutralizing titers after two vaccine doses received a third booster dose and enhanced the neutralization against numerous VOCs [95]. Neutralizing responses against Omicron in patients with solid cancer increased from 47.8% to 88.9% after a third-dose vaccination [86]. Enhanced immunogenicity was observed after a third mRNA COVID-19 vaccination in solid cancer patients who previously received the heterologous CoronaVac/ChAdOx1(inactivated vaccine) regimens [96]. In another study, nAbs in patients with solid cancer increased from 37% to 90% after the third dose against Omicron [53]. Hematological patients could also benefit from this process. In the same research, neutralizing antibodies against Omicron are rarely detected after two vaccine doses. However, approximately 50% have detectable neutralizing antibodies after a third dose [53]. Notably, booster vaccination might be especially important to patients with hematological malignancies, as they have a higher risk of not seroconverting after vaccination, especially in those with B cell malignancies receiving B cell-depleting therapies (CD20-targeted therapies or BTK inhibitors) [61,76,97].

Interestingly, the effect of prime–boost vaccination might be influenced by the vaccine combination or the sequence of injection. It seems that heterologous vaccination is superior to homologous vaccination, at least in those vaccinated with adenovirus-vectored vaccines as the first dose [98]. However, the conclusion was drawn from a small observational study. Thus, it may be too early to preclude any meaningful conclusions, not to mention the most effective vaccine combination.

Unfortunately, the effect of booster vaccination on T cell response remains unclear. A mild effect has been observed [98], while other studies indicate no significant increment [32] or discordant effects in patients who remain seronegative after booster vaccination [76].

## 4. Safety Concerns: Adverse Effects

The cancer population seems to exhibit similar rates of vaccine-induced adverse events compared to the general population. The most common adverse events reported in an early study involving patients with cancer were soreness or pain at or around the injection site (63% of vaccinees), local swelling (9%), and systemic reactions including muscle pain (34%), fatigue (34%), headache (16%), fever (10%), chills (10%), and gastrointestinal events (10%) [99]. A prospective observational study reported no evidence of other unknown adverse effects so far [20]. Because of short observation time, knowledge of longer-term adverse effects of COVID-19 vaccines in individuals with malignancy is limited.

Surprisingly, it has turned out that immune checkpoint inhibitors do not significantly increase immune-related adverse events (irAEs) rate [99,100,101]. This is in line with the VOICE trial; irAEs only occurred in 4% (6/137) patients treated with immunotherapy and 4% (7/163) patients treated with chemoimmunotherapy among participants in cohorts observed up to 28 days after the second vaccination [40]. However, possible vaccine-related irAEs are still reported in case reports [102,103].

Patients receiving hematopoietic stem cell transplantation (HSCT) or allogeneic hematopoietic cell transplantation (allo-HCT) seem to exhibit not only a reduced vaccine response, but also a greater risk of vaccine-induced adverse effects. Cytopenia and worsening of graft-versus-host disease may follow vaccination, and complications have been reported after vaccination in up to 10% of patients [71,82,104].

Local lymphadenopathy is commonly observed after vaccination. Breast cancer patients showed an 14.5% to 53% rate of vaccine-induced lymphadenopathy that lasted for over 6 weeks in 29% of patients [105]. Fortunately, lymphadenopathy usually resolves spontaneously [106]. However, the local lymphadenopathy brings challenges to imaging diagnoses, as it could be misinterpreted as lymph node metastases [44,105,107].

## 5. Discussion and Future Directions

Numerous studies have addressed the efficacy and immunogenicity of COVID-19 vaccines in patients with cancer since the outbreak of the pandemic. In this review, we provided information on vaccine effectiveness, factors that could influence the immune response, and vaccine safety of COVID-19 vaccines in patients with malignant disease. However, several questions remain, and additional research is required.

Firstly, studies are narrowed with the type of vaccine. Studies included in this review mostly focus on mRNA or adenovirus-vectored vaccines. Information on the efficacy and safety of other COVID-19 vaccines in patients with malignant disease is limited. For example, we currently have limited information on the inactivated vaccines used widely in China, a country with a large population of people living with malignant diseases.

Secondly, the endpoint of vaccine effectiveness varies among studies and should be considered deliberately. Most studies use the serological detection of spike-binding antibodies in patients with cancer as a major immunological endpoint. They assessed the presence of spike-reactive or RBD-reactive antibodies to establish rates of seroconversion and mean antibody titers. However, this may lead to imprecise implications of functional virus-neutralizing activity, especially against VOCs. Moreover, defining the levels of antibodies directed precisely against the RBD on the spike protein may be more clinically relevant as an indicator of protection against COVID-19 disease, as such antibodies have shown higher neutralization capacity. Previous studies are more abundant on the study of humoral immunity responses against COIVD-19 vaccines. However, studies on T cell responses have been addressed in relatively fewer studies and often in smaller subsets. This is probably because of the difficulty establishing risk factors and setting up a clear endpoint compared to humoral immunity. T cell positivity is defined variously among studies; some of them have measured T cell reactivity by broadly quantifying cellular cytokine secretion, while others have used the presence of certain activation markers on virus-specific T cells for this purpose. The role of the T cell response against COVID-19 should be paid more attention to, as T cell responses have been observed to be more robust in patients with hematological malignancies. Future research into T cell response may lead to more effective vaccines.

Thirdly, the stratification for specific cancer subtypes and therapies is needed. The importance of stratified subtypes has already been shown in the studies of hematological neoplasms. It is difficult to deduce the most optimal strategy of vaccination due to the heterogeneity of existing cohorts, while the efficacy of a vaccine may alter between different neoplasms subtypes. For example, differences in seroconversion of patients with malignant disease vs the general population are probably restricted to specific subgroups, highlighting the need for ongoing systematic meta-analyses to precisely define the at-risk groups among patients with either solid tumors or hematologic diseases. Future research should consider stratification in cancer subtypes when evaluating the safety and effectiveness of existing or novel COVID-19 vaccines. Moreover, questions of the optimal time between doses, the identification of at-risk patients after vaccination, and strategies for additional protection of at-risk patients beyond vaccination for different cancers are needed to be answered.

Fourthly, studies may be limited to publication time, therefore failing to include novel VOCs as study objects. Novel VOCs may perform different patterns of immune response after vaccination. For example, the progressive reduction in vaccine efficacy has been observed in new emerging VOCs, especially against the Beta, Delta, and Omicron VOCs. This means a portion of previous studies are time efficient. Novel studies are needed to answer the question of whether vaccine-induced protection could prevail against emerging VOCs, and whether novel vaccines are needed against later strains for better protection.

Finally, the prime–boost vaccination concept has already been adopted in many healthcare systems to provide a level of protection comparable to the general population. However, the question of whether heterologous vaccine boosting regimens generate improved vaccine-induced immune responses in patients with malignant disease needs further long-term evaluation. Additionally, the optimal combination of vaccines needs to be sought. Moreover, further investigations into the durability of vaccination and the therapy-specific influence over the vaccine needs to be launched.

## 6. Conclusions

Encouragingly, a massive reduction in the risk of severe COVID-19 and death has already been reached through global effort with the development of COVID-19 vaccines. The vaccines available are effective in patients with cancer and the incidence of severe adverse events is comparable to that in the general population, although patients with cancer seem more likely to develop a less proficient immune response than the general population. Certain toxicity profiles have been reported in specific patient populations, but the benefits of vaccination against COVID-19 clearly outweigh the risks in all patients with cancer.

A high proportion of patients with solid tumors can develop both humoral and T cell responses following vaccination, although cancer therapies can suppress these responses. Patients with hematological malignancies are often more vulnerable and perform limited immune responses; this is found in patients receiving high-dose steroids, chemotherapy, and especially in those with B cell malignancies receiving B cell-depleting therapies. Choosing a time when immunosuppressive treatments are withheld might be a strategy to improve VE.

Discouragingly, a proportion of patients exhibit seronegative responses. However, booster vaccines can result in seroconversion in those who were previously seronegative following two vaccine doses, suggesting benefits of continuing administered vaccine doses. In some cases, a third vaccine dose is strongly encouraged, particularly for individuals with hematological malignancies in whom two doses do not provide sufficient immune protection. A superiority has been observed in mRNA vaccines over viral vector vaccines regarding immunogenicity. More research into the combination of vaccine boosting is needed.

Effective measures taken to protect patients with cancer from contracting COVID-19 should not only include the prioritization of vaccination but also public health measures. Adequate vaccination of all close contacts to offer population immunity for cancer patients is needed.

## Figures and Tables

**Figure 1 vaccines-11-00486-f001:**
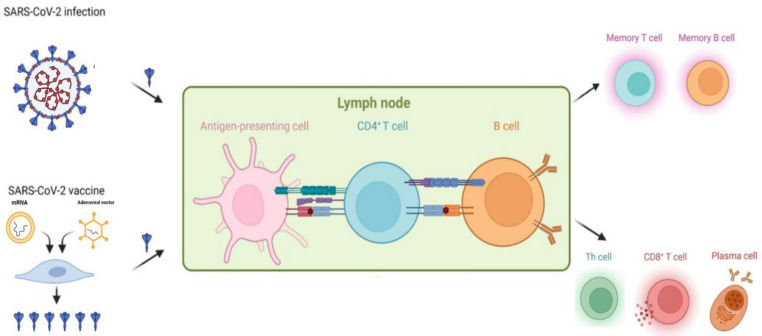
Infection- or vaccination-induced immune protection against SARS-CoV-2.

## Data Availability

Not applicable.

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
