# Peer review of "Effectiveness and Safety of COVID-19 Vaccination in Patients with Malignant Disease"

_vaccines, 2023, doi:10.3390/vaccines11020486_

Round 1

Reviewer 1 Report

This article is very interesting, updated, dealing with subject (immunization), that is very important for patients with malignancies.

Some English editing is required.

Author Response

Thank you for your review! Some english editing has been accomplished

Reviewer 2 Report

I read this review manuscript with great interest.

This review summarizes the current knowledge on the efficacy and safety of COVID-19 vaccine in patients with malignant diseases.

The issues are discussed in an organized manner for both solid tumors and hematologic malignancies. We believe that readers will find this review very informative.

Author Response

Thank you for your review! We made some English editing.

Reviewer 3 Report

The article COVID-19 vaccination in patients with malignant disease by Li Zhao et. al. claims to provide the immunological perspective of the two most used methods of immunization against COVID-19 infection.  The aims and objectives of the article have potential especially since it is very interesting to see the effect of immunization in patients who are on chemotherapy.

The authors fail to provide information regarding the intricacies involved in the process. The manuscript suggests no meaningful impact on B cell function, while some effect has been reported on T cells. It will be interesting to see which exact population of T cells is affected to a meaningful extent. 

There are several typos and incorrect sentences in the manuscript (for example line 120). The text will be more meaningful and easy to read if three-four cartoons will be included to provide a clear picture of the immune function.

It might be important to specify the outcome of immunization in the background of diseases such as chronic lymphocytic leukemia (CLL), non-Hodgkin’s lymphoma  (NHL), and multiple myeloma (MM), chronic myeloid leukemia (CML), myeloproliferative neoplasms (MPN), and myelodysplastic syndromes (MDS) perform different in seropositivity(CLL NHL and MM patients performed a much vulnerable immune response). Specific cancer therapies may also have contributed to impaired immunogenicity induced by vaccination. Immune cell-depleting treatmentshumoral immunity response is often drastically reduced due to CD20, BCMA, CD38 targeted therapies or other B cell-depleting therapies.

Author Response

Thank you for your review!

We didn’t provide specific information for B cell function as we focused on the humoral immunity as a whole and very few information of B cell function vaccination in patients with cancer was available.

Here are two articles we found that mentioned B-cell immune memory function in patients with malignant disease:

-Aleman A., et al. Variable cellular responses to SARS-CoV-2 in fully vaccinated patients with multiple myeloma. Cancer Cell. 2021;39:1442–1444.

- Shroff R.T., et al. Immune responses to two and three doses of the BNT162b2 mRNA vaccine in adults with solid tumors. Nat. Med. 2021;27:2002–2011.

We are happy to add information of B-cell function according to these two articles.

Information of T cell response are also limited and mostly focused on hematological malignancies. One study on MM patients (mentioned above) provided information on CD4+ and CD8+ T cell responses. Another article of myeloproliferative neoplasms also provided information on CD4+ and CD8+ T cell responses (Harrington P., et al. Single dose of BNT162b2 mRNA vaccine against SARS-CoV-2 induces high frequency of neutralizing antibody and polyfunctional T-cell responses in patients with myeloproliferative neoplasms). We have included the main information in our article, we could go into more details if needed. There is also an article (Marasco V., et al. T-cell immune response after mRNA SARS-CoV-2 vaccines is frequently detected also in the absence of seroconversion in patients with lymphoid malignancies. Br. J. Haematol. 2022;196:548–558.) that mentioned T-cell immune response but did not specify the exact population.

We made some English editing, including the paragraph you have mentioned.

We have added one cartoon to make the immune function clearer.

Reviewer 4 Report

The present work was a review focusing on effectivness and risks factor of COVID-19 vaccination in patients with malignancies: the aim was to provide a synthesis of pros, cons and issues in this particular population, that could experience immune system alteration for both pathological and treatment reasons.

Paper form must be revised, since there are confunding paragraphs and not very clear sentences. For example, in the first phrase "the virus" is in contrast the "a new coronavirus".

A major concern is the complete absence of literature search strategy, that make impossible to understand how the authors found articles and how they included or excluded them from the review.

In addition, many studies were reprorted with poor discussion and the collective significance remained unclear.

The paper must be improved to propose significant, novel and clear contributions.

Author Response

Thank you for your review!

We made some English editing to make the paper clearer.

We did not provide a literature search strategy as this article is not a systematic review.

In our paper, we have included the main information we considered significant. If you could provide us with any specific feedback, we would be happy consider and act on them.

We edited this paragraph in the introduction part to make the collective significance clearer:

Patients with malignant disease have a higher mortality rate on SARS-CoV-2. Therefore, it is vital that we understand how we can more effectively prevent infection or reduce disease severity through vaccination. This review will provide an overview of the current re-search of COVID-19 vaccines and patients with malignant disease, including vaccine effectiveness, factors that could influence the immune response, and vaccine safety. Information mentioned in this review may provide us with a better perspective to guide our clinical recommendations and future vaccination programs.

Round 2

Reviewer 4 Report

Some improvement was made, even not significant.

Even not mandatory, the description of literature search approach, with inclusion and exclusion criteria, let the readers to understand what authors aimed to find and whay some article was excluded.

Author Response

Thank you for your reply.  We would like to add this paragraph into our article:

In this review, we used different electronic databases including PubMed, EMBASE and the first 20 pages of the Google Scholar. The time scale was from December 15, 2019, to October 15, 2022. The language of the articles that were reviewed was limited to English. Information retrieval used the combinations of the following medical subject headings and key words: COVID-19, SARS‐CoV‐2, variants, alpha, beta, gamma, delta, iota, epsilon, vaccine, safety, efficacy, side effects, effectiveness, clinical trial, observational study, randomized controlled study, vaccination, mRNA vaccine, adenovirus vector vaccine, subunit vaccine, inactivated vaccine, ChAdOx1 nCoV-19, Ad26.COV2.S, mRNA-1273, BNT162b1, BNT162b2, rAd26, rAd5, MF59-adjuvanted spike glycoprotein-clamp, immunogenicity, immunotherapy, neoplasms, cancer, hematologic neoplasms, lymphoma, leukemia, multiple myeloma, myelodysplastic syndromes, myelofibrosis, B-Lymphocytes, T-Lymphocytes. Literatures were screened and we included relatively larger studies that described the immunogenicity, efficacy, and safety of COVID-19 vaccines in patients with cancer. Literatures were excluded with relatively small cohorts, except for areas that have limited articles.
